# Phylo-geo-network and haplogroup analysis of 611 novel coronavirus (SARS-CoV-2) genomes from India

Rezwanuzzaman Laskar , Safdar Ali

The novel coronavirus (SARS-CoV-2) from Wuhan China discovered in December 2019 has since developed into a global epidemic. Presently, we constructed and analyzed the phylo-geo-network of SARS-CoV-2 genomes from across India to understand the viral evolution in the country. A total of 611 full-length genomes from different states of India were extracted from the EpiCov repository of GISAID initiative on 6 June, 2020. Their alignment with the reference sequence (Wuhan, NCBI accession number NC_045512.2) uncovered 270 parsimony informative sites. Furthermore, 339 genomes were divided into 51 haplogroups. The network revealed the core haplogroup as that of reference sequence NC_045512.2 (Haplogroup A1) with 157 identical sequences present across 16 states. Remaining haplogroups had <10 identical sequences across a maximum of three states. Some states with fewer samples had more haplogroups. Forty-one haplogroups were localized exclusively to any one state. The two most common lineages are B6 and B1 (Pangolin) whereas clade A2a (Covidex) appears to be the most predominant in India. Because the pandemic is still emerging, the observations need to be monitored.

## Introduction

Coronaviruses belonging to the family Coronaviridae have been named so owing to the resemblance of the virion electron microscopic structure to that of a crown wherein the spikes present on the virion surface provide for the crown-like similarity (Lin et al, 2005). Their genome has a positive single strand RNA of 26–32 kb in length and are known to infect a wide range of hosts (Lai & Cavanagh, 1997; Ismail et al, 2003; Weiss & Navas-Martin, 2005; Cavanagh, 2007; Su et al, 2016).

The novel coronavirus SARS-CoV-2 from Wuhan, China, was discovered in December 2019. Since its emergence, it has developed into a global epidemic (Rothan & Byrareddy, 2020). As of 22 January, 2021, there were 10,625,428 cases and 1,53,032 deaths in India due to SARS-CoV-2 (https://www.mygov.in/covid-19). At the same time, as per World Health Organization there have been 96,012,792 confirmed cases including 2,075,870 deaths worldwide due to COVID-19 (covid19.who.int). The SARS-CoV-2 is different from earlier coronavirus outbreaks, severe

acute respiratory syndrome (SARS) coronavirus in 2002 and Middle East respiratory syndrome coronavirus in 2012 predominantly due to its extremely high transmission rates (Peiris et al, 2004; Zaki et al, 2012; Sun et al, 2020). The patients infected with SARS-CoV-2 have been observed to have varied symptoms ranging from normal flu like symptoms to high fever to invasive lesions (Chan et al, 2020; Huang et al, 2020; Zhu et al, 2020).

The SARS-CoV-2 belongs to genus betacoronavirus and subgenus sarbecovirus with possible origin in bats supported by its similarity to two bat-derived coronavirus strains, bat-SL-CoVZC45 and bat-SL-CoVZXC21 (Lu et al, 2020; Zhou et al, 2020). Also, the ever-increasing number of people being infected globally provides for the most conducive environments for the virus to evolve. The availability of full genome sequences for SARS-CoV-2 in Global Initiative on Sharing All Influenza Data (GISAID) has aided the study of these evolving sequences with both global and local perspectives (Shu & McCauley, 2017).

At present, we build and analyze the phylo-geo-network of SARS-CoV-2 based on the publicly available full-length sequences of SARS-CoV-2 from India. We also performed the haplogroup analysis with their defining mutations and phylogenetic lineage study along with geographical distributions. The present study would help us understand the evolutionary path of the virus in India, which would be an asset to counter the global burden of SARS-CoV-2 in future.

## Results

### Phylogenetic network analysis

The alignment of 611 SARS-CoV-2 genomes and their subsequent analysis revealed a total of 493 segregating sites of which 270 were parsimony informative (PI) sites. The incidence of sites and their distribution across gaps and ambiguous sequences and statistical evaluation has been summarized in Table 1. A negative value of Tajimas D statistic suggests the significance of these sites in evolution of these genomes. The reported phylo-geo-network herein has been built using the 152 PI sites obtained after excluding the gaps

---

Clinical and Applied Genomics Laboratory, Department of Biological Sciences, Aliah University, Kolkata, India

Correspondence: safdar_mgl@live.in; ali@aliah.ac.in

**Table 1.  Some key statistical parameters observed in the study.**

| S No | Network type | Number of segregating sites | Number of parsimony-informative sites | | Nucleotide diversity | Tajima's D statistic |
|------|--------------|------------------------------|----------------------|-----------|----------------------|-----------------------|
| | | | Excluding[a] | Including[a] | | |
| 1 | Transitive consistency score | 493 | 152 | 270 | $\pi = 0.00120683$ | D = −1.82662 p (D axis −1.82662) = 0.982906 |

[a]Gaps and ambiguous/missing (details in Table S3).

and ambiguous sequences. The phylo-geo-network analysis of the studied genomes has been represented in Fig 1.

Interestingly, there was one sequence with genome id 458080 from Telangana, which was 100 percent identical to the Wuhan reference sequence (Tables S1 and S4). Although the absence of travel history for most of the studied patients and the sequences only being a partial representation of the total patients present makes the conclusion subjective, it does indicate about arrival of the virus directly from China to India.

### Haplogroup analysis and distribution

The network tree construction was accompanied by haplogroup determination of the studied genomes. The nodes representing haplogroups in phylo-geo-network in Fig 1 have been named as per accession number of the sequence defining the haplogroup. The nodal haplogroup represented by the Wuhan reference sequence NC_045512.2 has two maxima associated with it. The node has 157 sequences distributed across 16 states. The details of distribution of all identical sequences have been summarized in Fig 2A and Table S2.

We propose the nomenclature of the 51 observed haplogroups as per the path used to construct the network which has been explained with a couple of examples as follows. The haplogroup having NC_045512.2 was named A1 as the core of the network. From this cluster, many haplogroups emerged and so on. The haplogroup A1.1 (420544) is defined by five positions; 241 (C → T), 3037 (C → T), 4809 (C → T), 14408 (C → T), and 23403 (A → G). However, as we move to haplogroup A1.1.1 (420543), in addition to the above mutations, another one at position 8782 (C → T) is present which becomes the defining polymorphism for this haplogroup. Similarly, haplogroup A1.6 (435063) is defined by positions 241(C → T), 1059 (C → T), 3037 (C → T), 14408 (C → T), 23403 (A → G), and 25563 (G → T). Subsequently haplogroup A1.6.1 (444471) is characterized by mutation at positions 18877 (C → T) and 26735 (C → T) and haplogroup A1.6.1.1 by additional mutations at 22444 (C → T) and 28854 (C → T). The haplogroup lineage thus defined clearly indicates that A1 is the most prevalent one, whereas A1.6 is the most evolving one as it has the maximum number of steps going up to A1.6.1.1.1.4 reflective of five steps and stages of mutations/PI sites. The position of all the observed PI sites has been listed in Table 2 and Fig 2B and their details are summarized in Table S3. The haplogroup nomenclature has been listed in correlation with their genome IDs and location in Table 3. The observed PI sites reported in the study include most of the commonly reported sites from across the world besides some novel ones. However, we are not emphasizing on the novelty of sites because of the fast-changing scenario and rapidly emerging data.

### Lineage and subtype analysis

We also ascertained the lineage and subtype of the observed sequences through Pangolin and Covidex, respectively. Also, the distribution of lineages present in India across the world was assessed through Pangolin. The fact that phylogenetic lineage of SARS-CoV-2 genomes from India exhibits its incidence in diverse countries such as USA, Australia, UK, Singapore, China, and Turkey is reflective of the global nature of the pandemic. Most of it can be attributed to international air travel and diverse regulations across countries.

The three most common lineages in India as predicted by Pangolin are B6, B1, and B1.36, whereas clade A2a appears to be the most predominant one as predicted by Covidex (Fig 2C and Tables 3 and S4). The prevalence of these lineages can shift with increasing incidences and accumulating variations and hence requires regular monitoring. However, proper recording of both national and international travel history for all the patients will go a long way in unveiling the true path of viral evolution.

# Discussion

The analysis of SARS-CoV-2 sequences from India through phylo-geo-network was carried out with the intention of analyzing the evolution of SARS-CoV-2 along with the geographical context. Subsequent to the Multiple Sequence Alignment, the network was constructed using 152 PI sites excluding the ambiguous sequences to ensure that only those sites wherein there was no sequence ambiguity formed the basis of the network.

Several observations can be drawn from the phylo-geo-network of the studied genomes as shown in Fig 1. First, the core of the network with maximum genomes (157) is the node of reference sequence of SARS-CoV-2 from Wuhan, China, with NCBI accession number NC_045512.2. The fact that this accounts for over one fourth (25.7%) of the total studied sequences is a clear indication that in spite of many reported variations, the original SARS-CoV-2 genome continues to be the dominantly prevalent form. Though the variations are fast accumulating in the virus, it's the original one that still prevails, at least in the Indian context. Viral evolution is a dynamic and fast process but unless due selection advantage is offered, a new form would not take over.

Second, the distribution of sequences from across India (Fig 1) do not corroborate with the incidence scenarios but are a reflection of the ground level preparations and activity in getting the genomes sequenced. For instance, the under-representation of Maharashtra and Tamil Nadu in the present data set in-spite of being the two most affected states. However, assuming that the virus has an equal

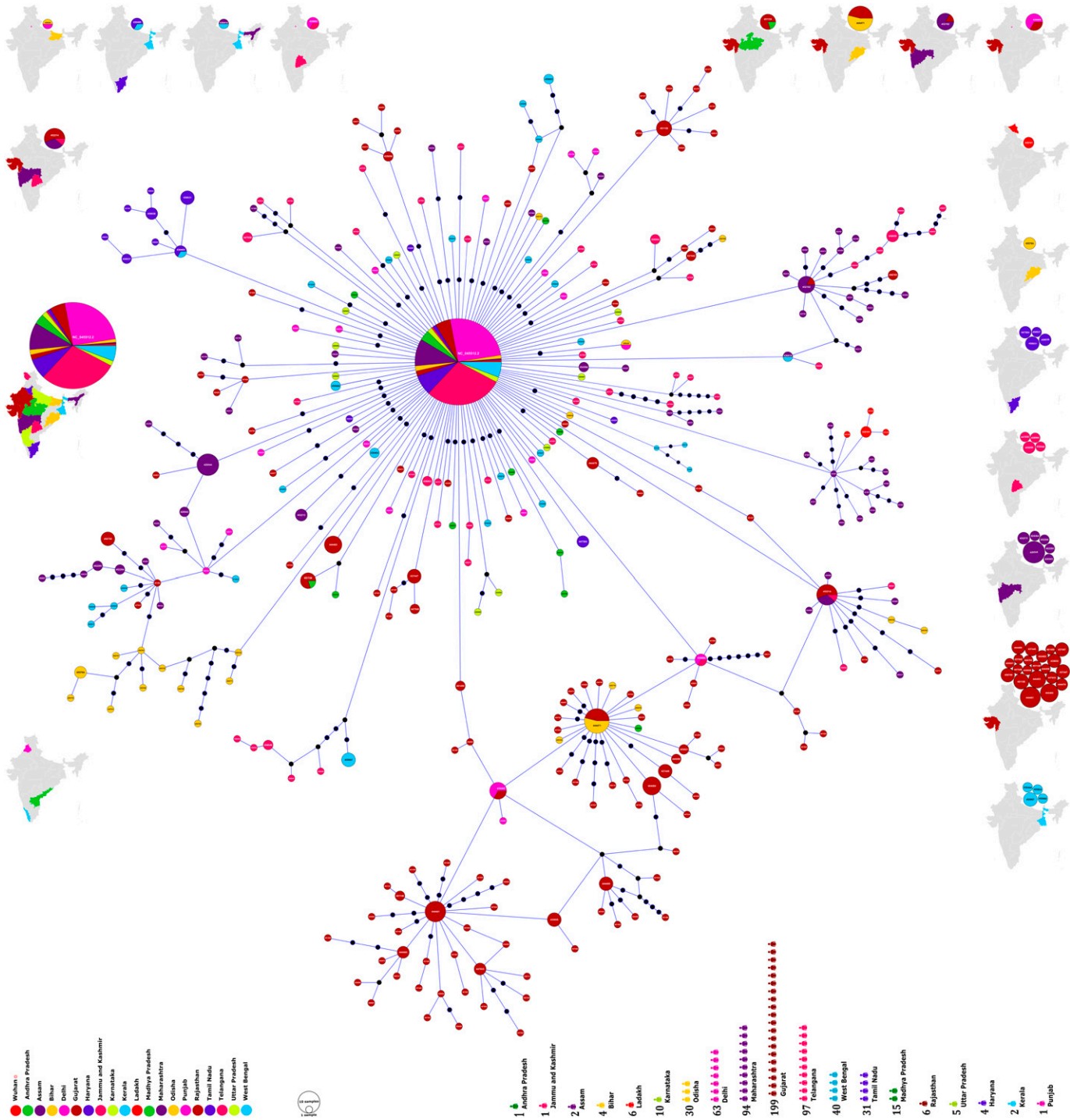

**Figure 1. Phylogenomic geographic (phylo-geo) network of SARS-CoV-2 genomes from India.**
The nodes represented by circles have been named after the Accession Numbers of the defining sequences representing a particular cluster. The diameter of the circle corresponds to the number of sequences present therein. Thus, a bigger circle will imply more sequences. The different states of India have been represented by color coding and the number of sequences from each state used in the study has been shown in the lower panel of the figure. The distribution of haplogroups across different states is shown in the maps on the periphery such that haplogroups present only in one state are in the maps on the right side. Maps on other sides include haplogroups present in more than one state. Maps have been generated and powered by Bing (Geo Names; Microsoft, TomTom) through MS Excel 2019.

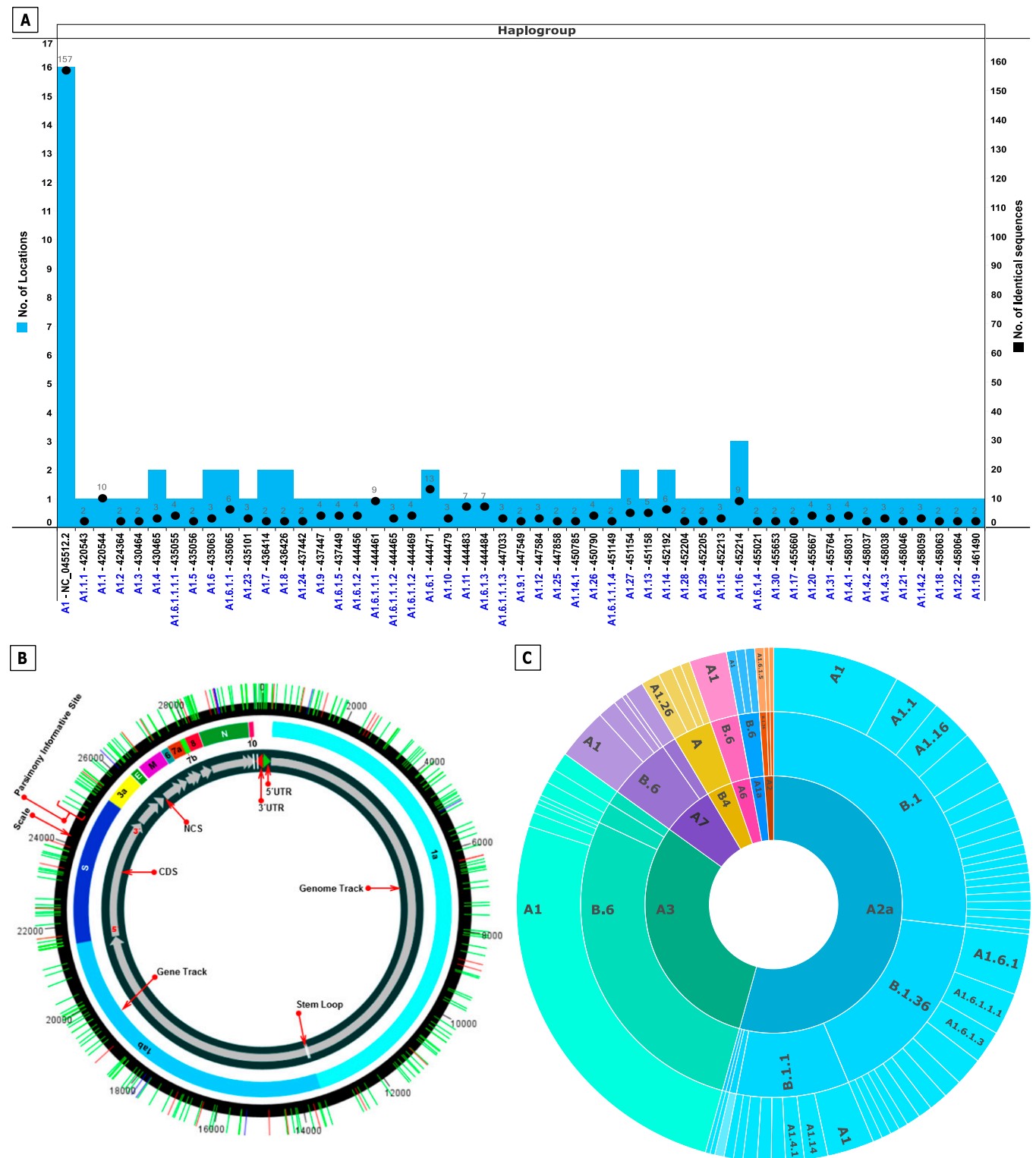

**Figure 2. Haplogroup distribution and lineage analysis of studied genomes. (A)** Prevalence and geographical distribution of 51 haplogroups of SARS-CoV-2 genomes in India. The haplogroups are shown on the x-axis. The number of identical sequences present in a haplogroup is shown as bar whereas number of states, wherein the haplogroup is present is shown as a black dot. Note the maximum prevalence (157 sequences) and widespread distribution (16 states) of NC_045512.2 containing haplogroup (A1). For details of haplogroup IDs, identical sequences, and locations, please refer Table S2. **(B)** Distribution of parsimony informative sites across the SARS-CoV-2 genomes. The SARS-CoV-2 genome has been represented circularly along with the locations of different genes/ORFs/Non coding regions. Parsimony informative sites are shown as lines traversing the circle. **(C)** Lineage and Subtype Analysis of SARS-CoV-2 genomes in India. The outermost circle represents haplogroups reported in the study whereas the middle circle depicts lineage prediction by Pangolin Web. The innermost circle is the clade analysis by Covidex Web tool.

**Table 2. Distribution of parsimony informative sites across the genome of nCOV-2019.**

| S No | Genome region | Start position | End position | Size (bp) | No of parsimony sites | Strike-rate of parsimony sites[a] | Position of parsimony informative sites including the gaps and ambiguous sequences |
|---|---|---|---|---|---|---|---|
| 1 | 5'UTR | 1 | 265 | 265 | 9 | 29.4 | 22, 55, 56, 94, 106, 218, 219, 222, 241 |
| 2 | ORF1a | 266 | 13,483 | 13,218 | 100 | 132.2 | 506, 635, 771, 875, 884, 1059, 1094, 1191, 1218, 1281, 1397, 1589, 1599, 1707, 1820, 1846, 2143, 2368, 2480, 2558, 2632, 2836, 3037, 3039, 3054, 3085, 3426, 3472, 3634, 3686, 3737, 3817, 4067, 4084, 4255, 4354, 4372, 4444, 4679, 4809, 4866, 4893, 4965, 5029, 5062, 5139, 5572, 5700, 5826, 6081, 6310, 6312, 6402, 6466, 6541, 6573, 6616, 6868, 6989, 7319, 7392, 7600, 7945, 8022, 8026, 8080, 8296, 8460, 8653, 8782, 8917, 8950, 9389, 9438, 9628, 9693, 10138, 10277, 10369, 10478, 10479, 10679, 10702, 10771, 10815, 11074, 11083, 11200, 11306, 11335, 11457, 11572, 11620, 12076, 12439, 12616, 12685, 12757, 13458 |
| 3 | ORF1b | 13,468 | 21,555 | 8,088 | 58 | 139.4 | 13585, 13617, 13730, 13859, 14130, 14181, 14274, 14408, 14425, 14673, 14805, 15324, 15435, 15451, 15708, 16017, 16078, 16355, 16393, 16626, 16738, 16852, 16887, 16993, 17135, 17440, 17722, 17747, 17858, 17959, 18052, 18129, 18380, 18395, 18457, 18486, 18511, 18877, 19086, 19185, 19344, 19417, 19524, 19679, 19684, 19816, 19872, 19983, 20006, 20063, 20087, 20151, 20355, 20773, 21004, 21137, 21550, 21551 |
| 4 | S | 21,563 | 25,384 | 3,822 | 33 | 115.8 | 21575, 21627, 21628, 21646, 21724, 21792, 21795, 21890, 22289, 22343, 22374, 22444, 22468, 22530, 22663, 23120, 23236, 23277, 23111, 23403, 23593, 23638, 23678, 23815, 23821, 23929, 24811, 24933, 25098, 25290, 25314, 25381 |
| 5 | ORF3a | 25,393 | 26,220 | 828 | 10 | 82.8 | 25445, 25461, 25513, 25528, 25563, 25596, 25613, 25855, 25904, 26144 |
| 6 | Non-coding | 26,221 | 26,244 | 24 | 1 | 24 | 26226 |
| 7 | E | 26,245 | 26,472 | 228 | 5 | 45.6 | 26330, 26338, 26375, 26376, 26467 |
| 8 | M | 26,523 | 27,191 | 669 | 6 | 111.5 | 26530, 26681, 26730, 26735, 27110, 27191 |
| 9 | ORF6 | 27,202 | 27,387 | 186 | 5 | 37.2 | 27213, 27379, 27382, 27383, 27384 |
| 10 | ORF7a | 27,394 | 27,759 | 366 | 1 | 366 | 27613 |
| 11 | ORF7b | 27,756 | 27,887 | 132 | 1 | 132 | 27874 |
| 12 | Non-coding | 27,888 | 27,893 | 6 | 1 | 6 | 27889 |
| 13 | ORF8 | 27,894 | 28,259 | 366 | 7 | 52.3 | 28001, 28077, 28083, 28114, 28221, 28253, 28254 |
| 14 | N | 28,274 | 29,533 | 1,260 | 20 | 63 | 28289, 28311, 28312, 28326, 28371, 28396, 28688, 28795, 28854, 28878, 28881, 28882, 28883, 28948, 29039, 29188, 29197, 29236, 29451, 29474 |
| 15 | Non-coding | 29,534 | 29,557 | 24 | 3 | 8 | 29543, 29555, 29557 |
| 16 | ORF10 | 29,558 | 29,674 | 117 | 0 | | |
| 17 | 3'UTR | 29,675 | 29,903 | 229 | 10 | 22.9 | 29722, 29734, 29742, 29743, 29774, 29827, 29829, 29830, 29870, 29874 |
| | Total | | | | 270 | | |

[a]Calculated by size/no. of parsimony sites in the region.

chance of evolving anywhere, we believe the number of sequences analyzed is apt for giving a glimpse of the ongoing viral evolution.

Third, when we analyzed the distribution of PI sites across the genome, we found it to be non-uniform in nature. We studied the distribution in the form of strike-rate of PI sites which we define as

the number of bases after which there will be another PI site. This is to say that a region with a strike rate of 20 would mean a PI site every 20 bases and so on. Thus, a lower strike rate will infer a higher density of the PI sites in the region (Table 2). Based on our analysis, the Envelope and Spike protein have a PI strike rate of 45 and 115,

**Table 3. Details of haplogroups: geographical distribution and phylogenetic lineage.**

| Haplogroup | Node label/ Genome ID | State | Most common countries | Lineage analysis (Rambaut et al, 2020 *Preprint*) | Subtype analysis-SARS Cov 2 nextstrain (Hadfield et al, 2018) |
|---|---|---|---|---|---|
| Proposed | Assigned by (GISAID/NCBI) | Assigned by Database (GISAID/NCBI) | Assigned by Pangolin Web server | Prediction by Pangolin Web server | Prediction by Covidex Web server |
| A1 | NC_045512.2 | 1. Assam | 1. Australia, Singapore, USA | 1. B | 1. A1a |
| | | 2. Bihar | 2. India, Singapore, Australia | 2. B.1 | 2. A2 |
| | | 3. Delhi | 3. UK, Australia, USA | 3. B.1.1 | 3. A2a |
| | | 4. Gujarat | 4. UK, China, USA | 4. B.1.5 | 4. A3 |
| | | 5. Haryana | 5. UK, Spain, Australia | 5. B.6 | 5. A6 |
| | | 6. Jammu | 6. UK, USA, Australia | | 6. A7 |
| | | 7. Karnataka | 7. UK, USA, China | | |
| | | 8. Madhya Pradesh | | | |
| | | 9. Maharashtra | | | |
| | | 10. Odisha | | | |
| | | 11. Rajasthan | | | |
| | | 12. Tamil Nadu | | | |
| | | 13. Telangana | | | |
| | | 14. Uttar Pradesh | | | |
| | | 15. West Bengal | | | |
| | | 16. Wuhan, China | | | |
| A1.1 | 420544 | Maharashtra | UK, USA, Australia | B.1 | A2a |
| A1.1.1 | 420543 | Maharashtra | UK, USA, Australia | B.1 | A2a |
| A1.10 | 444479 | Gujarat | UK, USA, Australia | B.1 | A2a |
| A1.11 | 444483 | Gujarat | UK, USA, Australia | B.1 | A2a |
| A1.12 | 447584 | Tamil Nadu | India, Singapore, Australia | B.6 | A3 |
| A1.13 | 451158 | Gujarat | UK, USA, Australia | B.1 | A2a |
| A1.14 | 452192 | 1. Gujarat | UK, Australia, USA | 1. B.1 | A2a |
| | | 2. Maharashtra | UK, USA, Australia | 2. B.1.1 | |
| A1.14.1 | 450785 | Gujarat | UK, USA, Australia | B.1 | A2a |
| A1.14.2 | 458059 | Telangana | UK, Australia, USA | B.1.1 | A2a |
| A1.15 | 452213 | Maharashtra | Australia, UK, Turkey | B.4 | A3 |
| A1.16 | 452214 | 1. Gujarat | UK, USA, Australia | B.1 | A2a |
| | | 2. Maharashtra | | | |
| | | 3. Telangana | | | |
| A1.17 | 455660 | West Bengal | UK, USA, Australia | B.1 | A2a |
| A1.18 | 458063 | Telangana | India, Singapore, Australia | B.6 | A7 |
| A1.19 | 461490 | Gujarat | UK, USA, Australia | B.1 | A2a |
| A1.2 | 424364 | Maharashtra | UK, USA, Australia | B.1 | A2a |
| A1.20 | 455667 | West Bengal | UK, USA, Australia | B.1 | A2a |
| A1.21 | 458046 | Telangana | UK, Australia, Gambia | B.1.1.8 | A2a |
| A1.22 | 458064 | Telangana | UK, USA, Australia | B.1 | A2a |
| A1.23 | 435101 | Ladakh | Australia, UK, Turkey | B.4 | A3 |

| Haplogroup | Node label/ Genome ID | State | Most common countries | Lineage analysis (Rambaut et al, 2020 *Preprint*) | Subtype analysis–SARS Cov 2 nextstrain (Hadfield et al, 2018) |
|---|---|---|---|---|---|
| A1.24 | 437442 | Gujarat | Australia, Singapore, USA | B.6 | A1a |
| A1.25 | 447858 | Telangana | India, Singapore, Australia | B.6 | A3 |
| A1.26 | 450790 | Gujarat | China, South Korea, USA | A | B4 |
| A1.27 | 451154 | 1. Gujarat | Australia, Singapore, USA | B.6 | 1. A3 |
| | | 2. Madhya Pradesh | India, Singapore, Australia | | 2. A7 |
| A1.28 | 452204 | Maharashtra | China, South Korea, USA | A | B4 |
| A1.29 | 452205 | Maharashtra | China, South Korea, USA | A | B4 |
| A1.3 | 430464 | West Bengal | UK, Australia, USA | B.1.1 | A2a |
| A1.30 | 455653 | West Bengal | UK, USA, Australia | B.1 | A2a |
| A1.31 | 455764 | Odisha | China, South Korea, USA | A | B4 |
| A1.4 | 430465 | 1. Tamil Nadu | UK, Australia, USA | B.1.1 | A2a |
| | | 2. West Bengal | | | |
| A1.4.1 | 458031 | Tamil Nadu | UK, Australia, USA | B.1.1 | A2a |
| A1.4.2 | 458037 | Tamil Nadu | UK, Australia, USA | B.1.1 | A2a |
| A1.4.3 | 458038 | Tamil Nadu | UK, Australia, USA | B.1.1 | A2a |
| A1.5 | 435056 | Gujarat | UK, USA, Australia | B.1 | A2a |
| A1.6 | 435063 | 1. Delhi | UK, USA, Australia | B.1 | A2a |
| | | 2. Telangana | | | |
| A1.6.1 | 444471 | 1. Gujarat | Saudi Arabia, UK, Turkey | B.1.36 | A2a |
| | | 2. Odisha | Turkey, Finland, UK | | |
| A1.6.1.1 | 435065 | 1. Delhi | Saudi Arabia, UK, Turkey | B.1.36 | A2a |
| | | 2. Gujarat | Turkey, Finland, UK | | |
| A1.6.1.1.1 | 444461 | Gujarat | Saudi Arabia, UK, Turkey | B.1.36 | A2a |
| | | | Turkey, Finland, UK | | |
| A1.6.1.1.1.1 | 435055 | Gujarat | Turkey, Finland, UK | B.1.36 | A2a |
| A1.6.1.1.1.2 | 444465 | Gujarat | Turkey, Finland, UK | B.1.36 | A2a |
| A1.6.1.1.1.3 | 447033 | Gujarat | Saudi Arabia, UK, Turkey | B.1.36 | A2a |
| | | | Turkey, Finland, UK | | |
| A1.6.1.1.1.4 | 451149 | Gujarat | Turkey, Finland, UK | B.1.36 | A2a |
| A1.6.1.1.2 | 444469 | Gujarat | Turkey, Finland, UK | B.1.36 | A2a |
| A1.6.1.2 | 444456 | Gujarat | Turkey, Finland, UK | B.1.36 | A2a |
| A1.6.1.3 | 444484 | Gujarat | Turkey, Finland, UK | B.1.36 | A2a |
| A1.6.1.4 | 455021 | Gujarat | Saudi Arabia, UK, Turkey | B.1.36 | A2a |

**Table 3.  Continued**

| Haplogroup | Node label/ Genome ID | State | Most common countries | Lineage analysis (Rambaut et al, 2020 *Preprint*) | Subtype analysis–SARS Cov 2 nextstrain (Hadfield et al, 2018) |
|---|---|---|---|---|---|
| A1.6.1.5 | 437449 | Gujarat | Turkey, Finland, UK | B.1.36 | 1. A2 |
| | | | | | 2. A2a |
| A1.7 | 436414 | 1. Assam | India, Singapore, Australia | B.6 | A1a |
| | | 2. West Bengal | | | |
| A1.8 | 436426 | 1. Bihar | India, Singapore, Australia | B.6 | 1. A3 |
| | | 2. Delhi | | | 2. A7 |
| A1.9 | 437447 | Gujarat | UK, USA, Australia | B.1 | A2a |
| A1.9.1 | 447549 | Gujarat | UK, USA, Australia | B.1 | A2a |

respectively (Table 2). Before drawing any conclusions, we need to understand that a higher incidence of PI sites does not necessarily corroborate to driving the evolutionary process as their impact on protein functionality needs to be ascertained first. However, it does indicate the potential genomic regions for the same which herein appear to be Envelope and Spike protein.

The geographical distribution of the haplogroups can be looked at from two different aspects. To begin with, which haplogroup is found in which location. Herein, A1 (NC_045512.2) haplogroup as already mentioned was most widely prevalent with 157 sequences distributed across 16 locations. All other haplogroups had 10 or fewer genomes spread across one to three locations (Fig 2A). The scenario is more interesting if we inverse the analysis as in which location had how many haplogroups. Gujarat with a maximal representation of 199 genomes had 27 different haplogroups but this is not the norm as in more sequences would mean more haplogroups. Delhi (63 genomes, 3 haplogroups), Maharashtra (94 genomes, 9 haplogroups), and West Bengal (40 genomes, 7 haplogroups) exhibit the non-linearity of the same. Also, 41 haplogroups have a single location only led by Gujarat (21); Maharashtra (6); West Bengal, Telangana, and Tamil Nadu (4 each); and Ladakh and Orissa (1 each). Three states, Punjab, Andhra Pradesh, and Kerala, do not have any haplogroup so far. The distribution of haplogroups across states has been shown in Fig 1 and Table S2. The fact that some locations with fewer samples have more haplogroups and most haplogroups are localized exclusively to a single state is a clear indication about the local evolution of viruses. However, because the pandemic is still emerging, the final outcome will be clear only at a later stage.

Of the 611 studied genomes, the 51 haplogroups account for 339 genomes. At this juncture, we would like to note about the sequences left out of haplogroups. They belong to haplotypes which may converge to an existing haplogroup or emerge as a new one as the pandemic progresses. Because of the high mutation rate of viruses and with ever increasing incidence of the diseases the virus is replicating more and more and new polymorphisms are being generated every day. These variations are changing the haplotype and haplogroup profile on a regular basis. The lineage and clade analysis of observed haplogroups was carried out through Pangolin and Covidex to correlate the network with global evolution of SARS-CoV-2. This was performed through Pangolin which analyses the evolution lineages and additionally reports their presence in

different areas of the world. The observed common lineages in India (B6, B1, and B1.36) and clade A2a as per present sequence congregation needs to be monitored regularly to understand the ongoing viral evolution.

## Conclusions

India provides for a good platform to understanding the emergence and evolution of SARS-CoV-2 pandemic because of its diseases burden spread over a huge and diverse population. The strain most prevalent in India is of the same haplogroup as the SARS-CoV-2 reference sequence from Wuhan indicating absence of any significant novel emerging strain so far. A total of 51 haplogroups have been reported. Geographical distribution of haplogroups across states and the corresponding number of genomes from the state suggest for a local evolution of the virus. The two most common lineages are B6 and B1 whereas clade A2a appears to be the most predominant one in Indian context. A regular update of the sequences and variations therein will help in deciphering SARS-CoV-2 evolution in India.

## Materials and Methods

### Sequence acquisition

Genome sequences of SARS-CoV-2 in FASTA format was assessed from the EpiCov repository (www.epicov.org) of GISAID initiative (Shu & McCauley, 2017) and reference sequence from Wuhan with accession number NC_045512.2 was retrieved from NCBI (www.ncbi.nlm.nih.gov).

On 6 June, 2020, we retrieved 611 FASTA sequence congregations along their rational meta data from GISAID EpiCoV server using the data filter ~ virus name: hCoV-19 - Host: Human - Location: Asia/ India – Complete – High Coverage and use the genome ID by excluding the first part i.e., "EPI_ISL_" of GISAID accession ID. Details of the geographical distribution of the sequences and their accession numbers are provided in Fig 1 and Table S1, respectively. Location data of GISAID are used to identify the state of origin in India, and wherein state name is unavailable, state address of the originating laboratory has been used. The workflow for the acquisition of sequences has been shown in Fig 3.

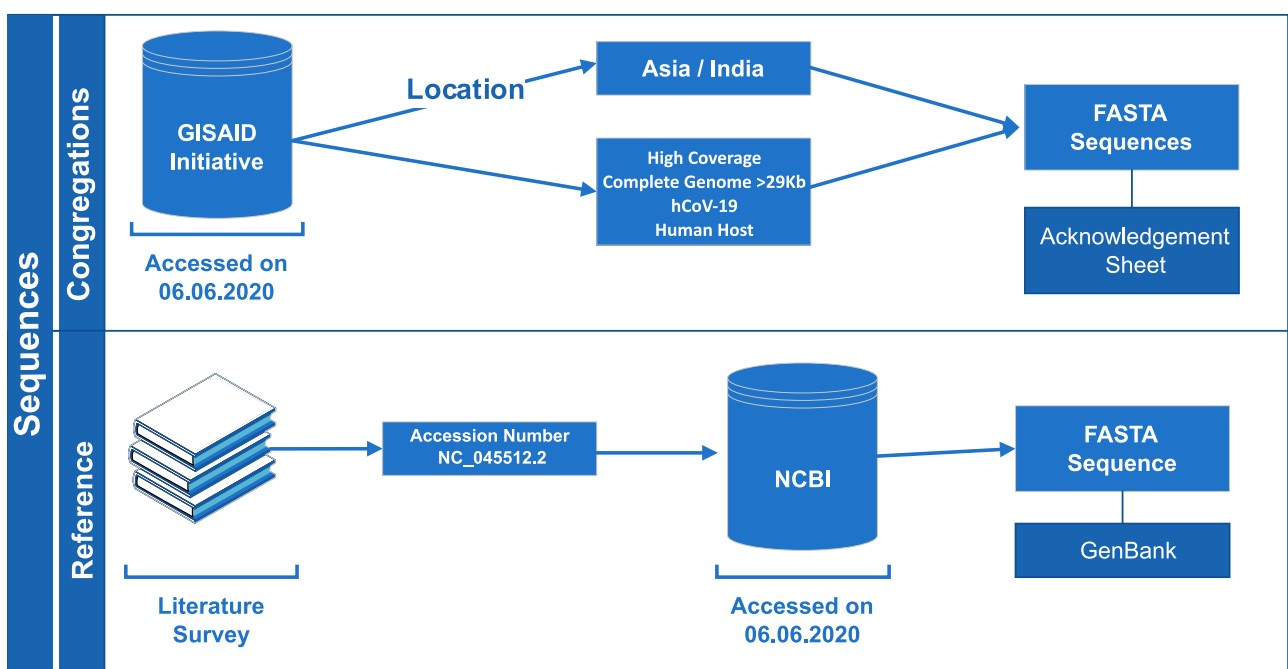

Figure 3.   Outline for selection and extraction of sequences used in the study.

## Sequence alignment

The congregations are aligned with the FFT-NS-fragment method using rapid calculation of full-length Multiple Sequence Alignment of closely related viral genomes, a light-weight algorithm of Multiple Alignment using Fast Fourier Transform v7 Web server (https://mafft.cbrc.jp/alignment/software/closelyrelatedviralgenomes.html) (Katoh et al, 2018) and keeping alignment size exactly throughout the reference sequence. The nucleotide transformation sites of the alignment were further studied using Molecular Evolutionary Genetics Analysis X (Kumar et al, 2018).

## Phylogenetic network analysis

Aligned sequences were used to generated parsimony based Transitive Consistency Score networks (Clement et al, 2002) implemented in Population Analysis with Reticulate Trees (PopART v1.7) software (Leigh & Bryant, 2015) where more than 5 percent sites contain undefined states and will be masked. A map of haplotypes was also drawn using the same software with geotags and traits label coding.

## Genome annotation

The tool Incorporation of Gene Location in SSR File (IGLSF) (Alam et al, 2019) arranges the location of variable sites according to genes. Using the software DNAPlotter (Carver et al, 2009), we used the Artemis (Carver et al, 2012) to annotate the genome and visualized it as a circular plot.

## Lineage and subtyping analysis

The global lineage to which the identified haplogroups from the sequence congregation belonged was ascertained through Pangolin (Phylogenetic Assignment of Named Global Outbreak Lineages) Web (https://pangolin.cog-uk.io/), using nomenclature implemented by Rambaut et al (2020) Preprint. Furthermore, the viral subtypes of the studied genomes from the Indian population was checked using "SARS Cov 2 Nextstrain" classification model of Covidex (https://cacciabue.shinyapps.io/shiny2/), a Web-based subtyping tool (Cacciabue et al, 2020 Preprint).

## Sequence statistics

Multiple metrics were used to assess the population genetics to decipher the phylogenetic relationship. We calculated Tajima's D (Tajima, 1989) statistic to test mutation–drift equilibrium and Π value, segregating sites, parsimony-informative sites to measure DNA polymorphism among sequences using PopART statistics (Leigh & Bryant, 2015).

# Data Availability

All data pertaining to the study has been provided as Supplementary Material of the manuscript.

# Supplementary Information

# Acknowledgements

The authors thank the Department of Biological Sciences, Aliah University, Kolkata, India, for all the financial and infrastructural support provided. The

authors acknowledge all the authors associated with originating and submitting laboratories of the sequences from GISAID's EpiFlu Database (www.gisaid.org) on which this research is based.

## Author Contributions

R Laskar: data curation, formal analysis, validation, and methodology.
S Ali: conceptualization, supervision, validation, and writing—original draft, review, and editing.

## Conflict of Interest Statement

The authors declare that they have no conflict of interest.

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
