## [Reviewer comments · Life Science Alliance]

Life Science Alliance

Phylo-geo-network and haplogroup analysis of 611 novel Coronavirus (SARS-CoV-2) genomes from India

Rezwanuzzaman Laskar and Safdar Ali

DOI: <https://10.26508/lsa.202000925>

Corresponding author(s): Safdar Ali, Aliah University

Review Timeline:	Submission Date:	2020-10-04
	Editorial Decision:	2021-01-21
	Revision Received:	2021-02-12
	Editorial Decision:	2021-03-01
	Revision Received:	2021-03-02
	Accepted:	2021-03-03

Scientific Editor: Shachi Bhatt

Transaction Report:

January 21, 2021

Re: Life Science Alliance manuscript #LSA-2020-00925-T

Safdar Ali
Aliah University
Biological Sciences
IIA/27 Newtown
Kolkata 700160
India

Dear Dr. Ali,

Thank you for submitting your manuscript entitled "Phylo-geo-network and haplogroup analysis of 611 novel Coronavirus (nCov-2019) genomes from India" to Life Science Alliance. The manuscript was assessed by expert reviewers, whose comments are appended to this letter.

We apologize for taking this long to get back to you. Unfortunately, it took us much longer than usual to fill a full panel of reviewers for this study, but ultimately we were able to secure 3 experts, who have now looked at this manuscript. All 3 reviewers find the study interesting and valuable, but they do have a number of requests, which we agree with. We would thus like to invite you to submit a revised version of your manuscript that addresses all of the reviewers' concerns.

Thank you for this interesting contribution to Life Science Alliance. We are looking forward to receiving your revised manuscript.

Sincerely,

Shachi Bhatt, Ph.D.
Executive Editor
Life Science Alliance
<https://www.lsjournal.org/>
Tweet @SciBhatt @LSAJournal

- A letter addressing the reviewers' comments point by point.
- An editable version of the final text (.DOC or .DOCX) is needed for copyediting (no PDFs).
- High-resolution figure, supplementary figure and video files uploaded as individual files: See our detailed guidelines for preparing your production-ready images, <https://www.life-science-alliance.org/authors>
- Summary blurb (enter in submission system): A short text summarizing in a single sentence the study (max. 200 characters including spaces). This text is used in conjunction with the titles of papers, hence should be informative and complementary to the title and running title. It should describe the context and significance of the findings for a general readership; it should be written in the present tense and refer to the work in the third person. Author names should not be mentioned.

B. MANUSCRIPT ORGANIZATION AND FORMATTING:

Reviewer #1 (Comments to the Authors (Required)):

Summary

This present study examined the phylogenomic network of nCov-2019 in India identifying the common haplogroups and lineage of the virus in the study population.

My major concern is the use of data generated since June 2020, having in mind the high mutation rate of SARS-CoV-2. More so, this manuscript is available as a preprint which was not highlighted to the reviewers. All headings were properly written with appropriate data, however, minor revisions

are required as stated below;

Kindly get the correct abbreviation of novel coronavirus (nCoV) in the title page and stick to it through the manuscript.

Abstract

Line 1

Author: The novel Coronavirus from Wuhan China discovered in December 2019 (nCoV-2019)

Reviewer: The novel Coronavirus (nCoV-2019) from Wuhan China discovered in December 2019.

Introduction

Page 3

Line 1 and 2: This is true. However, as this experiment was not carried out in the present study, a suitable and appropriate reference should be provided to substantiate this

Line 5: You can kindly include a reference to include the size of SARS-CoV-2 (nCoV-19 in Authors' words)

Line 9: Kindly update this statistic (28th August is a bit obsolete due to the increasing burden of COVID-19).

Line 10: It is pertinent to give the full meaning of abbreviations at first mention, kindly look into this as this was a common trend throughout the manuscript (and even the abstract). (NCBI, MSA, GISAID, WHO, etc)

Line 10-12: Revise this and you also need to update the data. You gave a statistic report from worldometer and acknowledged WHO. In case, you'd prefer WHO data; you can access that via covid19.who.int

Line 15-18: This should definitely be reviewed. How come articles published in 2004 (Peiris et al) and 2012 (Zaki et al) would serve as references for symptoms observed in nCoV-2019? Definitely not possible

Line 21-23: Somehow clumsy for me to understand. Revise this claim or provide a suitable reference

Page 4

Line 6-8: You are right about this. There is an increased burden of COVID-19 in India and most parts of the world. Nevertheless, I will appreciate optimism. With the heroic effort of researchers across the globe (including you), vaccines are being produced in large volumes. More so, despite the demography of India, adherence to preventive guidelines will go a long way in curtailing the menace of COVID-19.

Materials and Methods

Sequence acquisition

In the first sentence, you're stated to have mined sequences both from NCBI and GISAID but in the latter sentences, nothing was made mention of data filter on NCBI, how many sequences were gotten from NCBI and GISAID respectively, and the identifications of sequences mined from NCBI SARS-CoV-2 database. Likewise, the link you provided for NCBI is not the designated link for SARS-CoV-2. All these should be clearly stated for clarity and to improve the reproducibility of your method.

It is also a great concern for me since these 611 sequences were derived on 6th June 2020, which is already more than 6 months. I hope the claims would still be relevant as of when this study was done due to the ever-changing dynamics of the SARS-CoV-2.

Lineage and Subtyping Analysis

Kindly revise line 1 and 2 to aid clarity.

Results and Discussion

Phylogenetic network analysis

Line 1: The alignment of genomes (include number of genomes; 611 and name of the organism; nCoV-2019)

Page 7

Line 4 Can you clearly state accession no as accession number

Table 2

S/No 3 should refer to ORF1b and not ORF1ab, since S/No 2 already highlighted the PI sites in ORF1a

Under the genome region (column 2), since you are referring to base pairs and nucleotide positions, you should refer to SARS-CoV-2 genes instead of proteins. More so, ensure genes are in italics when this is revised.

We know of nucleocapsid, can you differentiate what are N and NC (as stated in column 2)? Kindly make this clear

Lineage and Subtype Analysis

Line 3-8: can you substantiate this with reference(s) highlighting the index cases of COVID-19 in India from the countries stated? This can as well serve as supporting information for the phylogenetic lineage

Conclusion

Line 1-2: Revise, this is not clear

Line 3: The strain or variant most prevalent in India is more appropriate

Reviewer #2 (Comments to the Authors (Required)):

In the manuscript "Phylo-geo-network and haplogroup analysis of 611 novel Coronavirus (nCoV-2019) genomes from India" Laskar and Ali analysed the phylo-geo-network of SARS-coV2 genomes to understand virus evolution in different geographical regions of India. The analysis of rapidly evolving viruses is very important to understand the evolution and geographical distribution of different virus variants. In this study, the authors extracted 611 full genomic sequences of SARS-coV2 from the different states of India. First genomic sequence alignment leads to identify 270 parsimony informative sites. second network analysis discovered that reference sequence NC_045512.2(Wuhan, China) forms the core haplogroup with 157 identical sequences present across 16 states of India. Further, in the comparative analysis of haplogroups, the authors observed local evolution of sars-coV2 genomes. Lastly, the data shows that B6 and B1 are the two most common lineages whereas the strains in A2a clade appears to be the most predominant in India.

Comments:

1. Indian territories are very diverse in terms of geographical conditions. Are differences in the haplogroups distribution in different states somehow linked to varying geographical conditions or is/are there some other reasons.
2. Does heterogeneity in haplogroups distribution in different states depends on the number of sequences analysed from each state? It would be interesting to know the distribution if same number of genomes are analysed from each state.
3. A variant of SARS-CoV-2 with a D614G mutation in the gene encoding the spike protein emerged in the beginning of 2020. After a couple of months, the D614G variant became dominant over initial SARS-CoV-2 strain originally identified in Wuhan, China. Have the authors detected the evolution/mutation of D614G spike variant in India? If yes, what is the level of distribution of D614G variants/mutants in different states of India?
4. Recently, a new variant of SARS-CoV2- called as VUI 202012/01, has been identified through viral genomic sequencing in the United Kingdom (UK). Its genome harbours multiple mutations (deletion 69-70, deletion 144, N501Y, A570D, P681H, T716I, S982A, D1118H) in the spike coding gene. Genomic sequence analysis revealed that currently the increase in SARS-coV2 cases in UK are associated with the VUI 202012/01 variant. Now, this VUI 202012/01 SARS-CoV2 variant is not

only present in UK but also small numbers of cases detected in other countries including in India. It would be intriguing to know what haplogroup this variant belongs to and I suggest the authors to include this data in the revised manuscript.

The findings are a novel contribution to the existing knowledge about Phylo-geo-network analysis of SARS-coV2 genomes across the different states of India. Overall, the present manuscript is well conceived, planned and executed. However, there are few minor concerns which must be addressed to further improve the quality of the manuscript.

Reviewer #3 (Comments to the Authors (Required)):

Overall, I find this paper to be an interesting addition to the current COVID-19 literature, especially as it focuses on India. Importantly this paper highlights local viral evolution and low overall genome evolution in relation to the Wuhan Genome.

However, I have the following comments.

1. The time when the genomes were obtained and analysed should be emphasized.
2. It should be emphasized that some of the language is a tad too simplistic in some paragraphs. For example, some lines of the abstract, the introduction and the results/discussion sections.
3. Some sentences are not clear in both context and structure. There are also minor grammatical errors and tense mistakes that create some confusion with understanding the work that was done.
4. The legend for figure 1 needs to be clearer, including the grammar. Actually, all figure legends should be rewritten to be clearer and easier to follow.
5. An explanation of the rationale behind the choice of methods is lacking. To fix this, I suggest that the result and discussion sections be separated, and rationale behind methods be explained in more depth in the discussion section.

Authors Response [AR] to Editor Comments [EC] and Reviewers Comments [RC] for Life Science Alliance manuscript #LSA-2020-00925-T entitled "Phylo-geo-network and haplogroup analysis of 611 novel Coronavirus (nCov-2019) genomes from India"

Editor Comments [EC]	Authors Response [AR]
A letter addressing the reviewers' comments point by point.	Provided with the revised manuscript.
A letter addressing the reviewers' comments point by point.	Provided with the revised manuscript.
An editable version of the final text (.DOC or .DOCX) is needed for copyediting (no PDFs).	Provided with the revised manuscript.
High-resolution figure, supplementary figure and video files uploaded as individual files	Provided with the revised manuscript.
Summary blurb (enter in submission system): A short text summarizing in a single sentence the study (max. 200 characters including spaces). This text is used in conjunction with the titles of papers, hence should be informative and complementary to the title and running title. It should describe the context and significance of the findings for a general readership; it should be written in the present tense and refer to the work in the third person. Author names should not be mentioned.	Provided with the revised manuscript.

Reviewer #1:

Reviewers Comments [RC]	[AR]
This present study examined the phylogenomic network of nCov-2019 in India identifying the common haplogroups and lineage of the virus in the study population. My major concern is the use of data generated since June 2020, having in mind the high mutation rate of SARS-CoV-2. More so, this manuscript is available as a preprint which was not highlighted to the reviewers. All headings were properly written with appropriate data.	The manuscript was available on preprint server since Sep 3 2020 at doi.org/10.1101/2020.09.03.281774, after which it was submitted to other journal which recommended the transfer to Life Science Alliance where it has been under consideration till date. Since, it was directly sent to journal and transferred therein we assumed the information about availability on preprint was also passed on. We sincerely regret the inconvenience caused. We agree with the accrual of mutations in SARS-CoV-2 and would be updating the data presented herein as short report/update as per journal norms at a later stage.
Kindly get the correct abbreviation of novel coronavirus (nCoV) in the title page and stick to it through the manuscript.	SARS-CoV-2 has been used throughout the revised manuscript.
Abstract Line 1 Author: The novel Coronavirus from Wuhan China discovered in December 2019 (nCOV-2019) Reviewer: The novel Coronavirus (nCOV-2019) from Wuhan China discovered in December 2019.	Revised accordingly.
Introduction Page 3 Line 1 and 2: This is true. However, as this experiment was not carried out in	Reference has been provided in revised manuscript.

the present study, a suitable and appropriate reference should be provided to substantiate this	
Line 5: You can kindly include a reference to include the size of SARS-CoV-2 (nCoV-19 in Authors' words)	Reference has been provided in revised manuscript.
Line 9: Kindly update this statistic (28th August is a bit obsolete due to the increasing burden of COVID-19).	Data has been updated in revised manuscript.
Line 10: It is pertinent to give the full meaning of abbreviations at first mention, kindly look into this as this was a common trend throughout the manuscript (and even the abstract). (NCBI, MSA, GISAID, WHO, etc)	Abbreviations list has been included in the revised manuscript.
Line 10-12: Revise this and you also need to update the data. You gave a statistic report from worldometer and acknowledged WHO. In case, you'd prefer WHO data; you can access that via covid19.who.int	Updated in the revised manuscript.
Line 15-18: This should definitely be reviewed. How come articles published in 2004 (Peiris et al) and 2012 (Zaki et al) would serve as references for symptoms observed in nCoV-2019? Definitely not possible	References for two statements were given together which led to the confusion. The said references are for previous incidences of SARS and MERS. The positioning of the references has been changed accordingly in the revised manuscript.
Line 21-23: Somehow clumsy for me to understand. Revise this claim or provide a suitable reference	Edited in the revised manuscript.
Page 4 Line 6-8: You are right about this. There is an increased burden of	We totally agree with being optimistic and have revised the statement accordingly.

COVID-19 in India and most parts of the world. Nevertheless, I will appreciate optimism. With the heroic effort of researchers across the globe (including you), vaccines are being produced in large volumes. More so, despite the demography of India, adherence to preventive guidelines will go a long way in curtailing the menace of COVID-19.	
Materials and Methods Sequence acquisition In the first sentence, you're stated to have mined sequences both from NCBI and GISAID but in the latter sentences, nothing was made mention of data filter on NCBI, how many sequences were gotten from NCBI and GISAID respectively, and the identifications of sequences mined from NCBI SARS-CoV-2 database. Likewise, the link you provided for NCBI is not the designated link for SARS-CoV-2. All these should be clearly stated for clarity and to improve the reproducibility of your method.	The genome congregation used for the study was extracted from GISAID as per parameters mentioned in methods section. NCBI was used for getting only the reference sequence from Wuhan. The same has now been clearly mentioned in the methods section and a workflow figure (Figure 3) for sequence extraction has also been provided in the revised manuscript.
It is also a great concern for me since these 611 sequences were derived on 6th June 2020, which is already more than 6 months. I hope the claims would still be relevant as of when this study was done due to the ever-changing dynamics of the SARS-CoV-2.	We agree with constant the accrual of mutations in SARS-CoV-2 and would be updating the data presented herein as short report/update as per journal norms at a later stage.
Lineage and Subtyping Analysis Kindly revise line 1 and 2 to aid clarity.	Edited in the revised manuscript.

Results and Discussion Phylogenetic network analysis Line 1: The alignment of genomes (include number of genomes; 611 and name of the organism; nCoV-2019)	Edited in the revised manuscript.
Page 7 Line 4 Can you clearly state accession no as accession number	Edited in the revised manuscript.
Table 2 S/No 3 should refer to ORF1b and not ORF1ab, since S/No 2 already highlighted the PI sites in ORF1a Under the genome region (column 2), since you are referring to base pairs and nucleotide positions, you should refer to SARS-CoV-2 genes instead of proteins. More so, ensure genes are in italics when this is revised. We know of nucleocapsid, can you differentiate what are N and NC (as stated in column 2)? Kindly make this clear	Edited in the revised manuscript.
Lineage and Subtype Analysis Line 3-8: can you substantiate this with reference(s) highlighting the index cases of COVID-19 in India from the countries stated? This can as well serve as supporting information for the phylogenetic lineage	The identification of index cases wasn't feasible due to absence of travel history for the studied sequences. Hence, the similarity of haplogroups with global lineage has been done using Pangolin which additionally monitors the presence of these haplogroups in different areas of the world as shown and mentioned in Table 3.
Conclusion Line 1-2: Revise, this is not clear Line 3: The strain or variant most prevalent in India is more appropriate	Edited in the revised manuscript.

Reviewer #2:

Reviewers Comments [RC]	Authors Response [AR]
In the manuscript ... to be the most predominant in India.	We thank the reviewer for a positive summary of our work.
1. Indian territories are very diverse in terms of geographical conditions. Are differences in the haplogroups distribution in different states somehow linked to varying geographical conditions or is/are there some other reasons.	Though we have observed and discussed the distribution/restriction of haplogroups across different states it would be slightly pre-emptive on our part to link it to geographical conditions at this stage because there is a very unequal distribution of samples from different states. This can be attributed more to socio-economic status than geography. States which have more international travel access like Maharashtra and Delhi have shown more cases than others. However, our group is under the process of studying and comparing data from respective states to ascertain possible geographical correlations, if any.
2. Does heterogeneity in haplogroups distribution in different states depends on the number of sequences analysed from each state? It would be interesting to know the distribution if same number of genomes are analysed from each state.	There is no uniform correlation between heterogeneity in haplogroup distribution and number of samples from a state as has been discussed in study as well. Delhi (63 genomes, 3 haplogroups), Maharashtra (94 genomes, 9 haplogroups) and West Bengal (40 genomes, 7 haplogroups) exhibit the non-linearity of the same. Also, there are haplogroups present in a single location: Gujarat (21), Maharashtra (6), West Bengal, Telangana, Tamil Nadu (4 each) and Ladakh, Orissa (1 each).

	The aspect of analyzing same number of sequences from each state isn't feasible herein as all states are not contributing equally to the disease incidence.
3. A variant of SARS-CoV-2 with a D614G mutation in the gene encoding the spike protein emerged in the beginning of 2020. After a couple of months, the D614G variant became dominant over initial SARS-CoV-2 strain originally identified in Wuhan, China. Have the authors detected the evolution/mutation of D614G spike variant in India? If yes, what is the level of distribution of D614G variants/mutants in different states of India?	Subsequent to the alignment of sequences while analysing our data using MEGA X we had an option of including/excluding the gaps and ambiguous sequences. Our analysis is based on 152 PI sites observed after excluding gaps and ambiguous sequences which doesn't include D614G mutation which in our alignment was present as ambiguous sequences. Since, we have based this study excluding ambiguous sequences, hence, D614G is not represented.
4. Recently, a new variant of SARS-CoV2- called as VUI 202012/01, has been identified through viral genomic sequencing in the United Kingdom (UK). Its genome harbours multiple mutations (deletion 69-70, deletion 144, N501Y, A570D, P681H, T716I, S982A, D1118H) in the spike coding gene. Genomic sequence analysis revealed that currently the increase in SARS-coV2 cases in UK are associated with the VUI 202012/01 variant. Now, this VUI 202012/01 SARS-CoV2 variant is not only present in UK but also small numbers of cases detected in other countries including in India. It would be intriguing to know what haplogroup this	There were 16 available sequences for the new variant available from India as accessed on 22/01/2021 but they were all of low coverage. However, since we have used only high coverage sequences in our original congregation hence a merger of the new data in this manuscript wasn't feasible. However, the new variant sequences from India represent three new haplogroups which is a part of an ongoing independent study of our group.

variant belongs to and I suggest the authors to include this data in the revised manuscript.	
The findings are a novel contribution to the existing knowledge about Phylo-geo-network analysis of SARS-coV2 genomes across the different states of India. Overall, the present manuscript is well conceived, planned and executed. However, there are few minor concerns which must be addressed to further improve the quality of the manuscript.	We thank the reviewer for the positive remarks and have addressed all the issues raised.

Reviewer #3:

Reviewers Comments [RC]	Authors Response [AR]
Overall, I find this paper to be an interesting addition to the current COVID-19 literature, especially as it focuses on India. Importantly this paper highlights local viral evolution and low overall genome evolution in relation to the Wuhan Genome.	We thank the reviewer for a positive summary of our work.
1. The time when the genomes were obtained and analysed should be emphasized.	The date of collection of sequences has been mentioned in the methods section and we have further added the analysis timeframe in details in the revised manuscript.
2. It should be emphasized that some of the language is a tad too simplistic in some paragraphs. For example, some lines of the abstract, the introduction and the results/discussion sections.	Edited in the revised manuscript.
3. Some sentences are not clear in both context and structure. There are also	Edited in the revised manuscript.

minor grammatical errors and tense mistakes that create some confusion with understanding the work that was done.	
4. The legend for figure 1 needs to be clearer, including the grammar. Actually, all figure legends should be rewritten to be clearer and easier to follow.	Legends have been revised accordingly.
5. An explanation of the rationale behind the choice of methods is lacking. To fix this, I suggest that the result and discussion sections be separated, and rationale behind methods be explained in more depth in the discussion section.	Results and Discussion are presented as separate sections in the revised manuscript and rationale behind methods included in discussion.

March 1, 2021

RE: Life Science Alliance Manuscript #LSA-2020-00925-TR

Dr. Safdar Ali
Aliah University
Biological Sciences
IIA/27 Newtown
Kolkata 700160
India

Dear Dr. Ali,

Thank you for submitting your revised manuscript entitled "Phylo-geo-network and haplogroup analysis of 611 novel Coronavirus (SARS-CoV-2) genomes from India". We would be happy to publish your paper in Life Science Alliance pending final revisions necessary to meet our formatting guidelines.

Along with the points listed below, please also attend to the following:

- Please use Capital Letters when introducing panels in Figure legends, e.g. instead of a) please use A
- please rename panels in Figure 2 as A, B, C (not as 2a, 2 b, and 2c) and correct callouts and figure legends accordingly
- please rename the datasets as supplementary tables and upload them in editable .doc or excel format

A. FINAL FILES:

B. MANUSCRIPT ORGANIZATION AND FORMATTING:

Sincerely,

Shachi Bhatt, Ph.D.

Executive Editor

Life Science Alliance

<https://www.lsjournal.org/>

Tweet @SciBhatt @LSAjournal

Interested in an editorial career? EMBO Solutions is hiring a Scientific Editor to join the international Life Science Alliance team. Find out more here -

https://www.embo.org/documents/jobs/Vacancy_Notice_Scientific_editor_LSA.pdf

Reviewer #1 (Comments to the Authors (Required)):

I commend the efforts of the Authors and Editor.

The manuscript has been revised and well updated. I recommend that it be considered for publication

Reviewer #2 (Comments to the Authors (Required)):

The findings are a novel contribution to the existing knowledge. In my opinion, this study not only helps us to understand the evolutionary path of the virus but also help in predicting the emergence of pathogenic mutations in the viral genome.

The authors have addressed the issues and in the revised version of the manuscript, all the main findings are fully supported by the data provided by the authors.

No additional comments. I recommend that the revised manuscript is suitable for publication in the Life Science Alliance journal.

March 3, 2021

RE: Life Science Alliance Manuscript #LSA-2020-00925-TRR

Dr. Safdar Ali
Aliah University
Biological Sciences
IIA/27 Newtown
Kolkata 700160
India

Dear Dr. Ali,

Thank you for submitting your Research Article entitled "Phylo-geo-network and haplogroup analysis of 611 novel Coronavirus (SARS-CoV-2) genomes from India". It is a pleasure to let you know that your manuscript is now accepted for publication in Life Science Alliance. Congratulations on this interesting work.

DISTRIBUTION OF MATERIALS:

Again, congratulations on a very nice paper. I hope you found the review process to be constructive and are pleased with how the manuscript was handled editorially. We look forward to future exciting submissions from your lab.

Sincerely,

Shachi Bhatt, Ph.D.

Executive Editor

Life Science Alliance

<https://www.lsajournal.org/>

Interested in an editorial career? EMBO Solutions is hiring a Scientific Editor to join the international Life Science Alliance team. Find out more here -

https://www.embo.org/documents/jobs/Vacancy_Notice_Scientific_editor_LSA.pdf